# The Role of TIM-3 in Glioblastoma Progression

**DOI:** 10.3390/cells14050346

**Published:** 2025-02-27

**Authors:** Farah Ahmady, Amit Sharma, Adrian A. Achuthan, George Kannourakis, Rodney B. Luwor

**Affiliations:** 1Fiona Elsey Cancer Research Institute, Ballarat, VIC 3350, Australia; farah@fecri.org.au (F.A.); george@fecri.org.au (G.K.); 2Federation University, Ballarat, VIC 3350, Australia; 3Department of Integrated Oncology, Center for Integrated Oncology (CIO) Bonn, University Hospital Bonn, 53127 Bonn, Germany; amit.sharma@ukbonn.de; 4Department of Neurosurgery, University Hospital Bonn, 53127 Bonn, Germany; 5Department of Medicine, The University of Melbourne, The Royal Melbourne Hospital, Parkville, VIC 3350, Australia; aaa@unimelb.edu.au; 6Department of Surgery, The University of Melbourne, The Royal Melbourne Hospital, Parkville, VIC 3350, Australia; 7Huagene Institute, Kecheng Science and Technology Park, Pukou District, Nanjing 211806, China

**Keywords:** TIM-3, glioblastoma, immunosuppression

## Abstract

Several immunoregulatory or immune checkpoint receptors including T cell immunoglobulin and mucin domain 3 (TIM-3) have been implicated in glioblastoma progression. Rigorous investigation over the last decade has elucidated TIM-3 as a key player in inhibiting immune cell activation and several key associated molecules have been identified both upstream and downstream that mediate immune cell dysfunction mechanistically. However, despite several reviews being published on other immune checkpoint molecules such as PD-1 and CTLA-4 in the glioblastoma setting, no such extensive review exists that specifically focuses on the role of TIM-3 in glioblastoma progression and immunosuppression. Here, we critically summarize the current literature regarding TIM-3 expression as a prognostic marker for glioblastoma, its expression profile on immune cells in glioblastoma patients and the exploration of anti-TIM-3 agents in glioblastoma pre-clinical models for potential clinical application.

## 1. Introduction

### 1.1. Glioblastoma Characteristics

Glioblastoma, classified as a grade IV, isocitrate dehydrogenase (IDH)-wildtype astrocytoma with at least one of the following features: microvascular proliferation, necrosis, EGFR gene amplification, TERT promoter mutation, or combined gain of chromosome 7/loss of chromosome 10 copy number changes, is the most frequent and aggressive adult malignancy in the central nervous system [1,2]. The current standard treatment for glioblastoma is the multimodality Stupp Protocol, involving surgical resection of the tumor mass followed by chemotherapy and radiotherapy treatments [3]. In this protocol, surgery precedes 6 weeks of adjuvant whole brain irradiation at 60 Gray (Gy) and concomitant and adjuvant temozolomide (TMZ) 75 mg/m^2^ both during and post-radiotherapy [4,5]. This treatment is followed by 30 days of daily irradiation at 2 Gy [4,5]. Although this is the standard worldwide treatment for glioblastoma, it has only increased overall survival (OS) by 2.5 months and has not significantly changed in over a decade [3,5,6,7,8]. This minor improvement may be due to glioblastoma’s highly invasive nature, widespread intra and intertumoral heterogeneity and resistance mechanisms, and differential drug penetration across the blood–brain barrier; as such, glioblastoma is currently considered incurable and prognosis is extremely poor, with a median OS of 12–15 months after diagnosis [7,9,10,11,12,13,14]. This is mainly due to invariable tumor recurrence arising in almost all glioblastoma patients [15,16,17]. Five-year OS remains at only 3–5%, 4.3% according to the Central Brain Tumor Registry, and ‘long-term survivors’ are classified as those who live longer than 2 years after diagnosis [5,18,19,20]. Glioblastoma is 1.6 times more likely to occur in males, older individuals (median age at diagnosis = 64 years) and those who have had previous irradiation therapy (particularly to the neck and head area), with familial inheritance only accounting for 1% of all glioblastomas [4,21,22,23,24,25].

### 1.2. Glioblastoma and Immunosuppression

It is well known that glioblastoma cells and the overall glioblastoma micro-environment have developed many immunosuppressive properties to protect themselves from immune cell recognition and attack. It is also clear that the ability of glioblastoma cells to evade or suppress immunosurveillance and response and ultimately block tumor cell elimination is a pivotal attribute contributing to the poor prognosis of glioblastoma patients. Several specific features favoring glioblastoma-mediated immunosuppression have been identified. One such tumor extrinsic mechanism involves the recruitment of T regulatory cells (Tregs) or myeloid-derived suppressor cells (MDSCs), which in turn secrete suppressive cytokines such as IL-10 and TGFβ, dampening immune responses. In 2006, two independent groups reported a significant increase in circulating Tregs in glioblastoma patients versus healthy controls [26,27]. Enhanced Treg frequency also correlates with reduced glioblastoma patient survival and time to recurrence [28,29]. Glioblastoma-associated microglia have also been observed to downregulate HLA and TNFα, while over-expressing immunosuppressive cytokines such as TGFβ and IL-10 [30,31,32].

However, the most studied immunosuppressive characteristic utilized intrinsically by glioblastoma tumors is the over-expression of immune checkpoint or inhibitory receptors. Programmed Cell Death 1 (PD-1; CD279) is the best characterized and studied immune checkpoint receptor in cancer, with PD-1 inhibitors showing strong antitumor activity in several cancer settings including melanoma and Non-Small Cell Lung Cancer (NSCLC) [33,34]. In contrast to other cancers, however, we and others have demonstrated that the expression of PD-1 or its ligand PD-L1 (CD274, B7-H1) are not independent prognostic markers for glioblastoma patients [35,36,37] and are relatively low in expression compared to other cancers [36,37,38]. These features of glioblastoma biology may partially explain why PD-1-based immunotherapy has failed to enhance survival in glioblastoma patients, along with other factors such as glioblastoma’s highly immunosuppressive microenvironment and/or poor T cell infiltration [39,40,41]. Together this suggests that PD-1 and PD-L1 may not play as large a role to the immunosuppressive landscape of glioblastoma compared to other cancer types and that potentially targeting other immunoregulatory receptors in the glioblastoma setting may be more beneficial. This review will focus on the less explored immunoregulatory receptor and exhaustion marker T cell immunoglobulin and mucin domain 3 (TIM-3) and its potential role in glioblastoma progression and immunosuppression. We will discuss studies describing TIM-3 as a potential biomarker for glioblastoma prognosis, its expression on several immune cell subtypes, and potential TIM-3 based therapeutic avenues that may offer greater clinical success than current immunotherapy-based agents.

## 2. TIM-3

### 2.1. TIM-3 Biology

Several key reviews have provided thorough historical and recent specifics of TIM-3 biology including its structure, function, and associated molecules related to regulation of TIM-3-induced immunosuppression and exhaustion [42,43,44,45]. As such, we will only provide a brief overview referencing key findings regarding TIM-3 biology. TIM-3, encoded by the *HAVCR2* gene was originally identified over two decades ago as a cell surface marker for interferon (IFN-γ) producing CD4^+^ and CD8^+^ T cells [46], and more recently expressed on Tregs, natural killer (NK) cells, macrophages, mast cells, and B cells (Figure 1) [47,48,49,50,51]. The inhibitory role of TIM-3 was first identified in an experimental autoimmune encephalomyelitis model of central nervous system autoimmunity [46].

Subsequently, TIM-3 was discovered in two back-to-back papers to play a key role in cancer immunosuppression and marks the most terminally dysfunctional or exhausted subset of CD8^+^ T cells [52,53], with expression on Tregs also leading to a highly suppressive environment [54,55]. Indeed, TIM-3 has been suggested to be a more reliable marker for terminally dysfunctional T cells compared to other immunoregulatory receptors such as PD-1 [52,53,56].

Molecularly, TIM-3 exerts its immune cell regulation through mechanisms including the blocking of T cell receptor (TCR) signaling and reduced association with HLA-B associated transcript 3 (BAT3; also known as BAG6). The association of TIM-3 and BAT3 is suggested to be dependent on the presence and biding of TIM-3 ligands. There are currently four identified ligands to TIM-3: galectin-9 (Gal-9), high mobility group protein B1 (HMGB1), carcinoembryonic antigen cell adhesion molecule 1 (CEACAM1), and phosphatidylserine (PtdSer) [57,58,59,60] (Figure 2). Without ligands, TIM-3 is bound to BAT3 at the cytoplasmic tail, rendering TIM-3 ineffective and allowing for enhanced T cell activity and cytotoxicity and enhanced secretion of inflammatory cytokines including IFNγ [61,62,63]. However, upon ligand binding, TIM-3 is released from BAT3 binding, leading to a cascade of sequential signaling events. These processes include the phosphorylation of TIM-3, BAT3 disassociation, potential enhanced TIM-3 binding of Fyn [64], phosphatase CD45 and CD148, and less BAT3/LCK association, which all leads to reduced TCR signaling [44,64,65]. Ligand binding to TIM-3 also triggers enhanced expression and transcriptional activity of B-lymphocyte-induced maturation protein 1 (BLIMP1) that mediates the regulation of several genes involved in T cell dysfunction and apoptosis [60,62,63].

Whether TIM-3 functional regulation occurs through similar signaling mechanisms in cells other than T cells is not well explored. In addition, despite research spanning two decades elucidating key features of TIM-3 biology, the role of TIM-3 in glioblastoma progression is not well studied. Very little is known regarding the transcriptional and epigenetic regulation of TIM-3 in the glioblastoma setting, although we have shown that T and NK cells from glioblastoma patients display reduced BAT3 expression, potentially allowing for enhanced TIM-3 expression and function [35]. As such, targeting TIM-3 in glioblastoma-based clinical trials has not been adequately evaluated. In the next section we will review the current literature regarding the potential supportive role of TIM-3 towards glioblastoma progression.

### 2.2. Correlating TIM-3 Expression and Glioblastoma Prognosis

Several studies, including our recent work [35], have demonstrated that the global expression of TIM-3 (or the *HAVCR2* gene that encodes for TIM-3) correlates with the survival of glioblastoma patients, and that *HAVCR2*/TIM-3 expression is significantly increased in glioblastoma compared to lower grade glioma and brain tissue that does not contain tumor [66,67,68,69,70,71,72,73,74,75]. A report by Guo and co-workers assessed the differential expression of 121 immune checkpoint markers between global glioblastoma and normal brain or low-grade glioma using the TCGA and GSE16011 glioma datasets [67]. The analyses identified only 7 out of 121 immune checkpoint markers (including *HAVCR2*/TIM-3) with elevated expression in glioblastoma compared to controls. Gene expression analysis of in-house glioblastoma samples confirmed that *HAVCR2* expression was higher than other well-known immune checkpoint inhibitors such as *CTLA4, PD-1, PD-L1, PD-L2,* and *IDO1*. Survival analysis with their in-house glioblastoma specimens also revealed that high TIM-3 protein expression indicated a shorter survival [67]. Finally, additional analysis demonstrated that TIM-3 protein expression was significantly higher in glioblastoma than that of grades II and III [74]. Another study evaluated the expression of *HAVCR2* (gene name for TIM-3) in different molecular subtypes utilizing the CGGA and TCGA databases and found that *HAVCR2* gene expression was the highest in mesenchymal glioma subtype [69]. In addition, glioma patients with the mesenchymal subtype exhibited significantly poorer survival to those patients with non-mesenchymal glioblastoma subtype [69]. A subsequent report that also analyzed the CGGA dataset and TCGA dataset along with the GSE16011 dataset, respectively, supported these earlier results, showing that *HAVCR2* gene expression was highest in the mesenchymal subgroup of glioblastoma patients and that these patients displayed shorter survival [72]. In addition, Han and colleagues showed that the percentage of TIM-3 positivity on peripheral CD8^+^ T cells (but not CD4^+^ T cells) inversely correlated with Karnofsky scores [68], while Liu et al. showed that the percentage of TIM-3 positivity on both tumor-infiltrating CD4^+^ and CD8^+^ T cells correlated with Karnofsky scores [71].

Several studies have evaluated the prognostic value of TIM-3 expression in combination with other potential markers or signatures. The DNA repair protein, O6 methylguanine-DNA methyltransferase (MGMT), directly inhibits the cytotoxic effect of temozolomide [76,77], and therefore patients with glioblastoma containing MGMT promoter methylation display better overall survival than those with functional and higher expression of MGMT [78,79,80,81]. Interestingly, a recent study identified that a subgroup of glioblastoma patients with high TIM-3 protein expression together with MGMT promoter non-methylation correlated strongly with shorter survival time [74]. However, whether MGMT methylation status and TIM-3 expression is linked mechanistically was not explored in this article and is currently not known. Although not as extensively studied in glioblastoma as MGMT methylation, the co-deletion of chromosome arms 1p and 19q (1p/19q co-deletion) is also observed in gliomas and in approximately 7–12% of glioblastomas [82,83]. It has been reported that patients with gliomas containing this co-deletion display a significantly greater survival time than those without the co-deletion [83,84,85], although another study showed no significant difference in glioblastoma patient survival between patients with the presence or absence of the 1p/19q co-deletion [82]. Nonetheless, a subsequent study indicated that the 1p/19q co-deletion in glioma can trigger a reduction in TIM-3 (and galectin-9) expression leading to enhanced antitumor activity, and thus providing a potential mechanism for why patients with this co-deletion may have enhanced survival [86]. The potential correlation between IDH mutation and TIM-3 expression has also been performed. In this study by Sorensen and colleagues, IDH mutation was associated with decreased levels of TIM-3 cells in an in-house cohort of glioblastoma and grade III glioma patient tissue and the TCGA database [87].

A study by Ni and colleagues demonstrated that patients with glioblastoma tumors that displayed high expression scores for galectin-9 and TIM-3 had significantly lower survival than patients with tumors with low expression scores of galectin-9 and TIM-3 [72]. Hu and co-workers evaluated the expression of TIM-3 together with the monocyte/macrophage marker CD68 in the TCGA, CGGA and their own in-house cohort for prognostic value. They demonstrated that patients with gliomas that had high TIM-3 and CD68 expression using the median expression as a threshold displayed significantly lower survival compared to glioma patients who expressed either high TIM-3/low CD68, low TIM-3/high CD68, or low TIM-3/low CD68 [88]. These findings were parallel with an earlier study where high CD68 expression correlated with poorer survival in glioma patients and robustly associated with expression of PD-1 and TIM-3 [89]. Similarly, two studies identified that high expression of the immune checkpoint protein CD96 correlated strongly with TIM-3 expression in glioblastoma samples and high expression of both was demonstrated to be associated with poor survival [90,91]. Similarly, CD204, a specific marker of tumor-associated macrophages (TAMs), was demonstrated to independently predict poor outcomes in glioma patients and its expression was closely associated with many immune checkpoint proteins including TIM-3 using the CGGA and TCGA dataset [92]. Another study showed that TIM-3 expression strongly correlated with CCL7, CCL18, and CXCL13 expression and that glioblastoma patients with high expression of these four markers have poorer survival outcomes [93]. Lin et al. evaluated TIM-3 expression with calcium related genes as a potential prognostic set for glioblastoma patient survival. Lin et al., created a calcium-related signature derived risk score based on high- and low-survival glioma patient groups and identified that this risk score associated with TIM-3 (and several other immune checkpoint proteins) [94]. Similarly, Wang and colleagues created an immune gene-related lncRNA risk model with the expression of the five LncRNAs identified strongly correlating with TIM-3 expression in glioma from the TCGA and CGGA databases [95]. Qi and colleagues demonstrated that an eight-gene risk signature based on fatty acid catabolic metabolism in glioblastoma had high prognostic value and that this signature significantly correlated with gene expression of the immune checkpoint markers *B7-H3* (CD276) and *HAVCR2* (TIM-3) [96]. Finally, Wang and colleagues demonstrated that the transmembrane protein 71 (TMEM71) gene was over-expressed in IDH-wildtype and MGMT-unmethylated patient samples and its expression was tightly associated with the gene expression of the PD-1, PD-L1, TIM-3, and B7-H3 [97].

However, despite obtaining information regarding which proteins are upregulated in correlation with upregulation of TIM-3, these studies did not explore whether these markers regulate each other’s expression and the potential pro-tumorigenicity-relevant to this correlative expression. In addition, in the correlative studies described above, it is unclear whether these TIM-3 expressing cells significantly contribute to the immunosuppressive phenotype of glioma/glioblastoma patients or are a byproduct of the tumor progression. These additional experiments would indeed be beneficial to understand the key molecular mechanisms that regulate TIM-3 expression and function, and which molecules/pathways are regulated by TIM-3 expression.

## 3. TIM-3 Expression on Immune Cells from Glioblastoma Patients

As outlined in Section 1.1, TIM-3 expression was originally identified on T cells and is predominantly studied for its role as a checkpoint or inhibitory receptor on several immune subsets. Several reports have evaluated the expression levels of TIM-3 on various immune cell sub-populations comparing expression on cells from glioma/glioblastoma tumor-infiltrating lymphocytes, peripheral blood mononuclear cells (PBMCs) from glioma/glioblastoma patients, and PBMCs from healthy donors. Han and colleagues compared TIM-3 expression on peripheral CD4^+^ and CD8^+^ T cells from 30 glioma patients of different grades and 30 healthy controls [68]. They showed that CD4^+^ and CD8^+^ T cells from high-grade glioma patients (grade III *n* = 9; grade IV *n* = 7) displayed significantly higher percentage of TIM-3 positivity than CD4^+^ and CD8^+^ T cells from low-grade glioma patients (grade I *n* = 5; grade II *n* = 9) [68]. Furthermore, they also identified that the percentage of TIM-3 positivity was significantly higher on peripheral CD4^+^ and CD8^+^ T cells from glioma patients overall (seven out of thirty were glioblastoma patients) compared to CD4^+^ and CD8^+^ T cells from healthy controls [68]. Similar results were obtained by Liu and colleagues. They also demonstrated greater percentage of TIM-3 positivity on peripheral CD4^+^ and CD8^+^ T cells from glioma patient tissues (58% of which were glioblastoma) compared to peripheral CD4^+^ and CD8^+^ T cells from healthy controls [71]. In addition, they and a more recent report from Fu and colleagues both showed that tumor-infiltrating CD4^+^ and CD8^+^ T cells expressed a significantly greater percentage of TIM-3 positivity compared to peripheral CD4^+^ and CD8^+^ T cells in matched glioma and glioblastoma samples respectively [71,98]. The study by Fu et al., however, did not present percentage of TIM-3 positivity comparisons of these groups with peripheral CD4^+^ and CD8^+^ T cells from healthy donors [98]. Another study also found a significant difference in the percentage of TIM-3 positivity of peripheral CD4^+^ T cells in glioblastoma patients compared to healthy controls [99]. Interestingly, in this study, the frequency of TIM-3 expression on CD4^+^ T cells in healthy donors was extremely low (below 1%). This study by Goods and colleagues also showed that the percentage of TIM-3 positivity on infiltrating CD4^+^ T cells was significantly higher than that on peripheral CD4^+^ T cells in glioblastoma patients. However, there was no significant difference between infiltrating and circulating CD4^+^ T cells in grade II and grade III glioma patients [99]. Meanwhile, Kim and colleagues observed that mice bearing the mouse glioma cell line GL261 had a higher percentage of TIM-3 positivity on brain-infiltrating CD4^+^ and CD8^+^ T cells compared to non-tumor–bearing control mice [100].

Two subsequent back-to-back reports published in Clinical Cancer Research also assessed TIM-3 expression on T cells from glioblastoma patients but showed contrasting results to the studies described above. The first by Woroniecka and colleagues evaluated TIM-3 expression on CD8^+^ T cells from healthy donor PBMCs, glioblastoma patient PBMCs, and tumor-infiltrating lymphocytes (TILs) [101]. This study demonstrated no significant difference in percentage of TIM-3 positivity in CD8^+^ T cells across all three groups. However, in mice orthotopically implanted with either CT2A or SMA-560 glioma cells, TILs displayed a significantly greater CD8^+^ T cell percentage of TIM-3 positivity compared to peripheral CD8^+^ T cells from glioma cell implanted mice and healthy mice [101]. The second study by Mohme and colleagues [17] also found no significant difference in percentage of TIM-3 positivity on peripheral CD4^+^ and CD8^+^ T cells comparing healthy donors, patients with newly diagnosed primary glioblastoma and recurrent glioblastoma. However, they did observe an increase in percentage of TIM-3 positivity on tumor-infiltrating CD4^+^ and CD8^+^ T cells compared to peripheral CD4^+^ and CD8^+^ T cells from both primary and recurrent glioblastoma patients [17]. Subsequently, Shen and colleagues also saw no significant difference in percentage of TIM-3 positivity on peripheral CD4^+^ and CD8^+^ T cells from glioma patients (13 out of 20 were glioblastoma) versus peripheral CD4^+^ and CD8^+^ T cells from healthy controls [102]. We recently observed similar findings where percentage of TIM-3 positivity on peripheral T cells (CD4^+^, CD8^+^, and Tregs) from glioblastoma patients were not significantly different to that seen on peripheral counterparts from age-matched healthy individuals [35]. Consistent with our data, Lowther et al., demonstrated that the percentage of TIM-3 positivity on peripheral Tregs from glioblastoma patients was not significantly different to peripheral Tregs from grade II and III glioma patients [103]. However, they showed that the percentage of TIM-3 positivity on infiltrating Tregs was significantly greater than that of peripheral Tregs from glioblastoma patients. This significant difference was not observed when comparing the percentage of TIM-3 positivity on tumor-infiltrating versus peripheral Tregs from grade II and grade III glioma patients [103]. It is difficult to determine the exact reasons for these large discrepancies in findings across these studies. However, potential explanations may include differences in patient cohorts, sample size, the storage conditions and length of samples, experimental technique differences, and variable tumor microenvironmental factors.

Others have examined the expression of TIM-3 on monocytes/macrophages in the glioblastoma setting. The study by Lehman and colleagues assessed the expression of several immune function and activity markers on three subsets of monocytes: classical (CD14^+^ CD16^−^ SLAN^−^), intermediate (CD14^+^ CD16^+^ SLAN^−^), and non-classical (CD14^low^/^−^ CD16^+^ SLAN^+^) from glioma patients (15 out of 24 were glioblastoma) and healthy individuals [104]. Despite the intermediate group expressing significantly higher levels of the anti-inflammatory cytokines TGFβ and IL-10, no significant change in percentage of TIM-3 positivity was observed between all three subgroups of monocytes from glioma patients versus healthy controls [104]. Interestingly, another group identified significantly greater percentage of TIM-3 positivity on peripheral CD14^+^ monocytes from glioma patients versus healthy donors and glioma patients with higher percentage of TIM-3 positivity on CD14^+^ monocytes having a higher risk for tumor recurrence or death [70]. The expression status of CD16 and SLAN was not determined in this study; however, they did show that the percentage of TIM-3 positivity on monocytes that also expressed CD206, a common marker for M2 macrophages, was significantly higher than on CD14^+^, CD206^-^ monocytes from glioma patients suggesting that the M2 macrophage phenotype express higher TIM-3 and pro-tumor properties [70]. A more recent study also determined that pro-tumor M2 macrophages express more TIM-3 compared to antitumor M1 macrophages in glioma patients [88]. These findings support a previous study where Ni et al. examined the role of PTEN in TIM-3 expression and M2 macrophage pro-tumorigenic function [72]. Mechanistically, they showed that PTEN null glioblastoma cells secreted higher levels of the TIM-3 ligand, galectin-9, leading to a TIM-3-dependent polarization of M2 macrophages. This subsequently resulted in enhanced VEGF-A expression and secretion from these M2 macrophages promoting glioblastoma tumor growth [72]. Another potential mechanism of enhanced TIM-3 frequency and the M2 macrophage phenotype was recently elucidated by Zhang and colleagues [75]. They showed that the nuclear factor of activated T cells-1 (NFAT1) upregulates TIM-3 expression in THP-1 monocyte cells via the enhanced secretion of complement 3a (C3a), which then bind to C3aR and promote the M2-like macrophage polarization by activating TIM-3. In addition, siRNA knockdown of TIM-3 led to reduced M2-like TAMs, suggesting that NFAT1 upregulated TIM-3 in TAMs, which further promoted M2-like polarization [75].

We and others have also examined the percentage of TIM-3 positivity on NK cells from glioblastoma patients. The study by Li and colleagues assessed the percentage of TIM-3 positivity on NK cells from 25 glioma patients and 17 healthy controls [70]. They found that glioma patients had a higher percentage of TIM-3 positivity on peripheral CD3^-^ CD56^+^ NK cells compared to NK cells from healthy donors. In addition, TIM-3^+^ NK cells secreted less IFNγ compared to TIM-3^-^ NK cells and glioma patients with higher percentage of TIM-3 positivity on their surface, and peripheral CD3^-^ CD56^+^ NK cells also had higher levels of proliferative tumor cells as measured by Ki-67 expression levels. However, the percentage of TIM-3 positivity on CD3^-^ CD56^+^ NK cells did not correlate with patient survival [70]. The study by Shen and colleagues demonstrated similar results, showing that NK cells from glioma patients (13 out of 20 were glioblastoma patients), had a significantly higher percentage of TIM-3 positivity compared to NK cells from healthy controls [102]. A study by Wang and colleagues successfully engineered dual-specific chimeric antigen receptor (CAR) that target both disialoganglioside (GD2) and ligands to NK group 2D (NKG2D) into the NK cell line NK92 and primary NK cells [105]. These engineered cells displayed significantly greater cytotoxicity towards glioblastoma cells compared to parental NK cell counterparts and significantly impaired glioblastoma xenograft growth in vivo. Interestingly, these engineered NK cells expressed lower TIM-3 compared to parental NK cells. However, whether the increased cytotoxicity was due to reduced TIM-3 expression was not determined in this study [105]. Another study did somewhat prove that TIM-3 expression on NK cells was important for anti-glioblastoma effects [106]. Morimoto and colleagues successfully knocked down TIM-3 in healthy human peripheral blood-derived NK cells and showed that these reduced TIM-3 expressing NK cells also had reduced ability to inhibit glioblastoma cell growth in vitro compared to control NK cells [106].

Finally, one study has assessed TIM-3 expression on natural killer T (NKT) cells in the glioma setting and found no significant difference in the percentage of TIM-3 positivity on NKT cells from glioma patients compared to NKT cells from healthy individuals [70]. A very recent publication identified enhanced percentage of TIM-3 positivity on tumor-infiltrating immune cells in both myeloid and lymphoid compartments from pilocytic astrocytoma (the most common pediatric glioma) patients compared with matched patient PBMCs and healthy control PBMCs [107]. Another study showed that exosomes from the cerebral spinal fluid of glioblastoma patient contained galectin-9, which could hinder antigen recognition, processing, and presentation by dendritic cells (DCs) in a TIM-3-dependent mechanism. This inhibition of DC function subsequently blocked cytotoxic T-cell-mediated antitumor immune response [108]. Malo et al. also observed that blocking VEGF led to a more mature DC phenotype in the brain and a reduction in TIM-3 expression on brain-infiltrating CD8^+^ T cells in a mouse glioma model [109].

Our search through the current literature did not yield any studies examining TIM-3 expression and potential immunoregulatory features driven by TIM-3 regarding B cells in the glioblastoma micro-environment. Nonetheless, the outlined studies in this section, taken together, suggest that TIM-3 expression on T cells, monocytes/macrophages, and NK cells plays a key role in glioblastoma induced immunosuppression. However, almost all these studies outlined in this section evaluated TIM-3 expression on the immune cell surface without stimulation, and thus did not evaluate if TIM-3 expression is potentially dysregulated in glioblastoma patients when cells are activated. We observed similar results to others where the percentage of TIM-3 positivity was not significantly different on unstimulated CD4^+^ and CD8^+^ T cells and NK cells from healthy donors and glioblastoma patients [35]. However, healthy donor CD4^+^ and CD8^+^ T cells stimulated with phorbol 12-myristate 13-acetate (PMA) and ionomycin had a significantly lower percentage of TIM-3 positivity compared to stimulated CD4^+^ and CD8^+^ T cells from glioblastoma patients [35]. These results suggest that some potential key differences in immune cell function may only be elucidated when cells are challenged and/or stimulated as would be the case in vivo in both healthy individuals and glioblastoma patients. Another advantage of examining changes with and without stimulation is the greater accuracy of determining changes of markers that indicate activity and in some cases cytotoxicity. Our recent work did not identify differences in the percentage of CD69 and IFNγ positivity in unstimulated peripheral CD4^+^ and CD8^+^ T cells and NK cells from newly diagnosed glioblastoma patients versus healthy donors potentially due to the expected low levels of CD69 and IFNγ positivity when cells were unstimulated. However, we saw a significant reduction in the percentage of CD69 and IFNγ positivity in PMA and ionomycin-stimulated CD4^+^ and CD8^+^ T cells and NK cells from glioblastoma patients compared to their counterparts in healthy donors [35]. Thus, research on immune cells with and without stimulation provides a greater overall landscape of potential global glioblastoma-driven immunosuppressive features occurring within patients.

## 4. TIM-3 Expression on Glioblastoma Cells

TIM-3 expression has been detected on cancer cells of various types, although the exact reason for TIM-3 expression and the potential signaling networks and regulation of gene expression based on this TIM-3 mediated signaling is not well explored [110,111,112,113,114,115]. An initial study by Kim and colleagues assessed TIM-3 expression in tumor tissue from eight primary glioblastoma patients by IHC [100]. They found that four of the eight glioblastoma tissue sections showed a staining pattern consistent with positive tumor cells [100]. Zhang and colleagues also showed that the U251 and U87 glioblastoma cell lines expressed more TIM-3 than 293T cells, while knockdown of TIM-3 sensitized these cells to temozolomide induced reduction in cell viability and enhanced apoptosis [116]. However, neither of these studies evaluated the molecular mechanisms that drive enhanced TIM-3 expression on glioblastoma cells. A more recent paper by Guo and colleagues performed a thorough assessment of a potential pro-tumorigenic role of TIM-3 expression on glioblastoma cells [67]. Examining previous single-cell RNA-seq data by Neftel et al. [117], they identified *HAVCR2* gene-expressing tumor cells and TIM-3 expression in primary tissue cell suspensions from clinical glioblastoma samples, and primary and established glioma cell lines [67]. Subsequent over-expression of TIM-3 in glioblastoma cell lines led to an NFκB/interleukin-6 (IL-6)-dependent enhancement of proliferation, migration, and invasion in vitro, and significant tumor growth and reduced mouse survival in vivo [67].

Glial cells consisting of oligodendrocytes, astrocytes, microglia, and ependymal cells have several essential functions in the brain. These include supporting neuron structure and function, providing growth factors and nutritional support, maintaining a functional blood–brain barrier, generating myelin, cell–cell communication, and contributing to the local immune response [118,119,120,121]. A recent study by Yuan and colleagues identified higher expression of TIM-3 on microglia in patient glioblastoma tissue compared to adjacent brain tissue using single-nucleus RNA sequencing and spatial transcriptomics analysis [122]. Kim et al. explored a potential role of TIM-3 expression on glial cells in a mouse brain tumor model [123]. They found that microglia, T cells, and tumor cells all expressed TIM-3 in brain sections of mice orthotopically implanted with GL26 glioma cells. However, gene expression of *HAVCR2* and TIM-3 promoter activity was rapidly and significantly reduced in mouse and rat primary microglia exposed to conditioned media from mouse, rat, and human glioma cell lines, and in glioma bearing mice in vivo. In addition, TIM-3 expression was required in glial cells to induce IFNγ expression from CD8^+^ T cells in co-culture experiments. These results are somewhat contradictory to the above literature, which depict TIM-3 as a pro-tumorigenic molecule mainly through promoting immune cell in-activity. We can only speculate that these disparity with the bulk of the current literature may be due to differences in mouse versus human glial behavior, the time points chosen where initial contact with tumor cells may evoke an immune response but later produce chronic inactivity or a context or cell type specific differential function that is currently not well understood. Nonetheless, this article by Kim et al. suggests that TIM-3 expression on glial cells may promote T cell activity through IFNγ expression [123]. Clearly, further elucidation of the exact interplay between tumor cells, glial cells, and immune cells within the glioblastoma micro-environment, the key roles and significance of this interplay in glioblastoma progression, and potential novel avenues of targeted inhibition are still required.

## 5. Targeting TIM-3 in Glioblastoma

Despite some clear evidence that TIM-3 may play a key immunoregulatory role that promotes glioblastoma development and progression, currently, clinical evaluation of anti-TIM-3 agents in the glioblastoma setting is limited. However, several pre-clinical studies have assessed the use of anti-TIM-3 agents or agents that potentially indirectly block the immunoregulatory function of TIM-3 in the glioblastoma setting. One of the first reports to describe anti-TIM-3 therapy in the glioblastoma pre-clinical setting was by Kim and colleagues in 2017 [100]. They showed that 250 μg/mouse of an anti-murine TIM-3 antibody given at day 7, 11, and 15 post intracranial implantation of GL261 mouse glioma cells had no significant effect to mouse median or overall survival (OS) compared to control treated mice. Similarly, monotherapy of anti-PD-1 antibody or stereotactic radiosurgery monotherapy or dual therapy of either combination of anti-TIM-3, anti-PD-1, or stereotactic radiosurgery treatment resulted in no or small sub-optimal increases in mouse survival. However, triple therapy of anti-TIM-3, anti-PD-1, or stereotactic radiosurgery treatment resulted in significantly greater survival compared to all other tested treatment groups (Figure 3) [100].

This improved survival from the triple therapy was significantly impaired when either anti-CD4 and anti-CD8 antibodies were injected to depleted mice of CD4^+^ or CD8^+^ cells, suggesting that the full effects of this antitumor triple therapy require functional CD4^+^ or CD8^+^ cells [100].

Li and colleagues using the same GL261 intracranial mouse model and the same anti-TIM-3 antibody (Clone RMT3-23) as Kim et al. [100], observed a small yet significant increase in mouse survival compared to control treated mice, when 250 μg/mouse of an anti-murine TIM-3 antibody was administered at day 12, 15, and 18 post tumor cell implantation [124]. Interestingly, mice bearing GL261 intracranial tumors treated with the combined therapy of the anti-TIM-3 antibody and an antibody targeting the TIM-3 ligand CEACAM produced a significantly greater survival than mice treated with control or either antibody as monotherapy. Similarly to Kim et al., this enhanced survival was diminished when anti-CD4 and anti-CD8 antibodies were also administered to the mice [124]. The disparity in efficacy of this anti-TIM-3 antibody across these two studies could be due to differences in the inoculation method, inoculation position, inoculation cell number, or scheduling of the anti-TIM-3 antibody administration. Another recent article, however, showed that the human anti-TIM-3 antibody (BMS-986258) could not significantly extend the survival of mice bearing CT-2A mouse glioma cells, but could significantly enhanced the survival of mice bearing diffuse intrinsic pontine glioma (DIPG) cells intracranially [125]. Similarly, anti-TIM-3 antibody treatment significantly enhanced the median survival of a genetically engineered murine model of MAPK-driven, low-grade gliomas, compared to mice treated with IgG control or an anti–PD-1 antibody. This therapeutic inhibition by the anti-TIM-3 antibody was diminished in mice with a CD8 knockout background [107]. Ni and colleagues also examined the effects of a murine anti-TIM-3 antibody in several rodent models. They showed that administration of 100 μg/mouse of the anti-TIM-3 antibody daily for 7 days resulted in significant increases in mouse survival in an orthotopic human U87MG glioblastoma model, a syngeneic GL261-PTEN knockout mouse model, and a syngeneic C6 rat model [72]. Interestingly, the anti-TIM-3 antibody inhibited the intracranial growth of human glioblastoma cell line U87MG in nude mice, suggesting that the antitumor effects of this anti-TIM-3 antibody may not be completely dependent on evoking or enhancing an immune response.

Finally, several articles have described other therapeutic approaches that lead to changes in TIM-3 expression on brain tumor-infiltrating T cells. OX40 (CD134; TNFRSF4) is a member of the tumor necrosis factor receptor family and is expressed on activated T cells, providing co-stimulatory signal for T cell activation [126,127,128]. A study by Jahan and colleagues assessed the efficacy of an anti-OX40 agonistic monoclonal antibody that stimulates CD4^+^ and CD8^+^ T cells in combination with a vaccine approach involving subcutaneously injected GL261 mouse glioma cells that over-express GM-CSF and have been irradiated prior to administration (GVAX) against the GL261 intracranial mouse glioma model [129]. The combination of this anti-OX40 antibody and GVAX resulted in significantly enhanced survival compared to both therapies as monotherapy and control treated mice. The combination therapy also significantly reduced TIM-3, PD-1, and LAG expression on intracranial infiltrating T cells [129]. Another report also showed that blockade of VEGF led to a reduction in TIM-3 expression on brain-infiltrating CD8^+^ T cells in a mouse glioma model [109]. Another novel combination therapy was evaluated by Wang and co-workers [130]. In this report, they demonstrated that the FDA approved anti-depressant agent paroxetine combined with biomimetic nanoparticles resulted in significantly reduced GL261 intracranial tumors and significantly enhanced survival compared to treatment with each agent as monotherapy or control treated mice. In addition, the combination treatment mediated a significant reduction in the percentage of TIM-3 positivity on tumor-infiltrating T cells [130]. In contrast, the most used chemotherapeutic to treat glioblastoma patients, temozolomide, when administered at a high dose schedule (50 mg/kg daily for 5 days) into mice bearing GL261 and KR158 murine glioma models led to an upregulation of exhaustion markers LAG-3 and TIM-3 on CD4^+^ and CD8^+^ T cells [131]. Importantly, treatment of mice with the same total amount of temozolomide but at half the dose for double the duration (25 mg/kg daily for 10 days) did not result in any increase in LAG-3 and TIM-3 on CD4^+^ and CD8^+^ T cells [131]. This report, together with future results describing temozolomide and its influence on the expression of TIM-3 and other immunoregulatory receptors, may led to modifications to the current standard management of patients with primary and/or recurrent glioblastoma.

## 6. Concluding Remarks

The discovery that tumor and immune cells express many immunoregulatory or checkpoint molecules that prevent immune response and cytotoxicity led to excitement that blockade of these molecules could promote antitumor activity and deliver improved patient survival outcomes. However, targeting the first wave of these molecules, such as PD-1 and CTLA-4, has yielded disappointing overall outcomes in the glioblastoma setting. As outlined here, many reports have identified TIM-3 as an important inhibitory receptor in the glioma/glioblastoma setting, although much remains to be elucidated regarding its signaling and regulatory networks that mechanistically control and dampen immune cell activity in glioblastoma. Nonetheless, the current body of work exploring the key role(s) of TIM-3 expression in patient prognosis and overall TIM-3 biology in the glioblastoma space has provided adequate evidence that therapeutic modulation of TIM-3 is worthy of investigation. Indeed, it is clear that TIM-3 expression on immune, glial, and glioblastoma cells correlates with worse prognosis. However, further illumination of the exact role of TIM-3-mediated immunosuppression and subsequent glioblastoma progression is required. Likewise, current pre-clinical models demonstrate that TIM-3 blockade can result in reduced glioblastoma growth and enhanced survival in vivo and combination therapies including TIM-3 inhibitors show greater promise than monotherapies. Further evaluation of anti-TIM-3 agents (potentially in combination with blockade of other inhibitory receptors such as PD-1 and CTLA-4) in glioblastoma should be strongly encouraged, with the aim that therapeutic regimens centered around blocking TIM-3 function will potentially improve glioblastoma patient survival.

## Figures and Tables

**Figure 1 cells-14-00346-f001:**
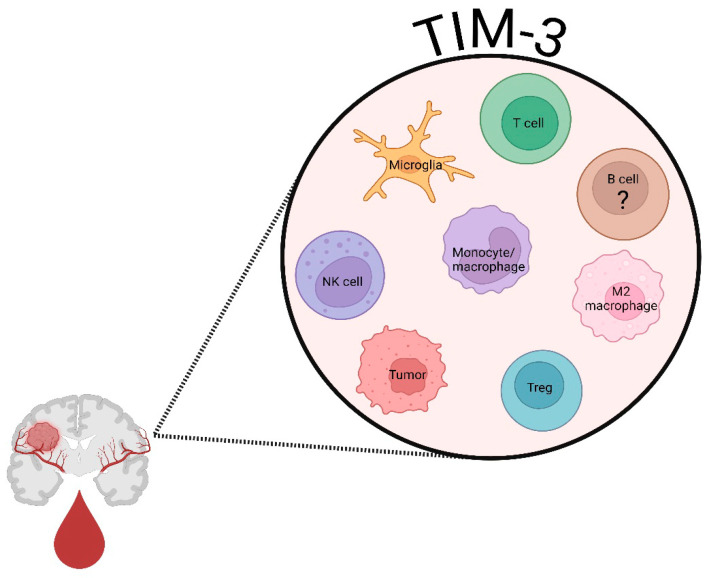
Cells which express TIM-3 in the glioblastoma tumor micro-environment. A range of immune cell subsets, including innate (i.e., NK cells, monocytes, and macrophages) and adaptive (B cells, T cells, and Tregs) cells, have been reported to express varying levels of TIM-3 on their surface in either/both the peripheral blood of glioblastoma patients and at the tumor site. Brain cells, such as microglia and glioblastoma tumor cells, also express varying levels of TIM-3 on their surface. Created in BioRender.

**Figure 2 cells-14-00346-f002:**
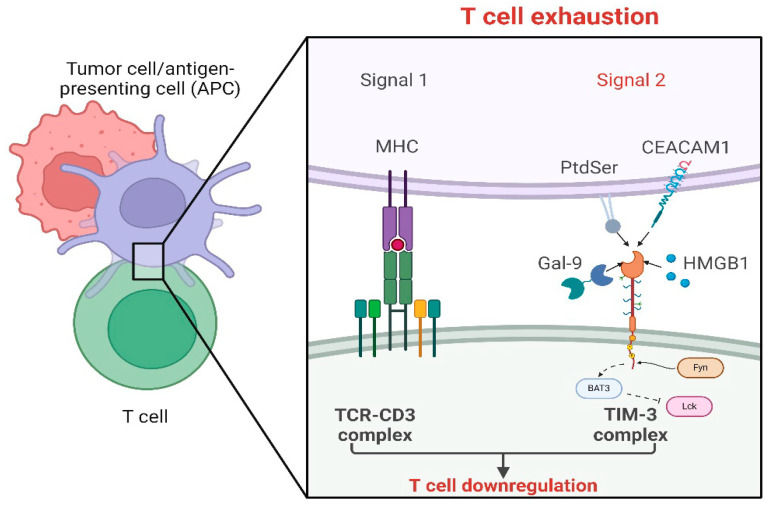
T cell exhaustion in a TIM-3-dependant manner. Mechanistic pathway leading to T cell exhaustion. Tumor cells and/or antigen presenting cells (APC) interact with the T cells with signal 1 consisting of antigen being presented by tumor and/or APC on their major histocompatibility complex to the T cell receptor (TCR) of the T cell. This interaction is a co-stimulatory response. Signal 2 consists of either cell surface TIM-3 ligands (i.e., phosphatidylserine (PtdSer) and carcinoembryonic antigen cell adhesion molecule 1 (CEACAM1) or soluble TIM-3 ligands (i.e., galectin-9 (Gal-9) and high mobility group protein B1 (HMGB1) binding to the TIM-3 receptor, initiating a signaling cascade. Binding of any of these ligands to the TIM-3 receptor causes the transcription factor BAT3 to be released from the cytoplasmic tail of the TIM-3 complex, leading to enhanced binding of Fyn instead, and leading to the reduction in the BAT3/Lck association. This process results in the downregulation of T cells and ultimately leads to T cell exhaustion. Created in BioRender.

**Figure 3 cells-14-00346-f003:**
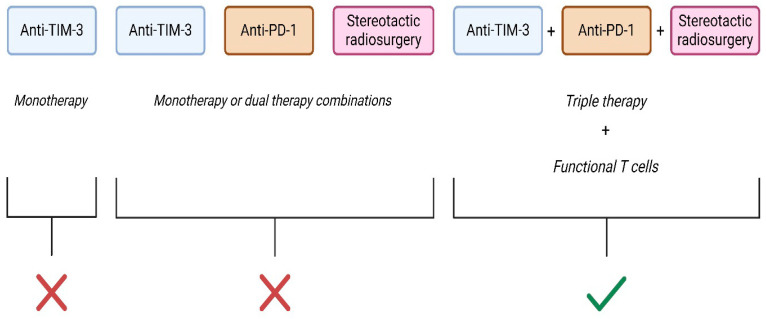
Therapy combinations and survival outcomes for glioblastoma patients. Monotherapy and/or dual therapy combinations (consisting of either anti-TIM-3, anti-PD-1, or stereotactic radiosurgery) has either no effect or small sub-optimal increases in survival in mice bearing glioblastoma tumors. Triple combination therapy consisting of anti-TIM-3, anti-PD-1, and stereotactic radiosurgery improves survival which is dependent on the presence of functional T cells (CD4+ and CD8+ T cells). Created in BioRender.

## Data Availability

Not applicable.

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
