# Peer review of "The Role of TIM-3 in Glioblastoma Progression"

_cells, 2025, doi:10.3390/cells14050346_

Round 1
Reviewer 1 Report
Comments and Suggestions for Authors
Comments
This review provides an in-depth discussion of the role of TIM-3 in glioblastoma progression, focusing on its function in immunosuppression and potential therapeutic interventions. The manuscript is well structured and comprehensive. The authors have effectively compiled and critically analyzed existing literature, highlighting gaps in knowledge and future research directions. However, certain aspects require clarification, additional discussion and refinement to enhance the manuscript’s clarity and impact. Below are specific comments and suggestions for improvement.
Glioblastoma characteristics
1. Authors have mentioned that Stupp protocol has increased survival by 2.5 months. Briefly explain the potential reason (such as glioblastoma’s resistance mechanisms, invasive nature, or limited drug penetration across the blood-brain barrier).
2. Previous brain irradiation is mentioned as a risk factor. A short clarification on whether this refers to prior therapeutic radiation (such as for other tumors) or environmental exposure should be given.
Glioblastoma and Immunosuppression
1. Authors have discussed glioblastoma’s immunosuppressive properties. Clearly differentiate between tumor-intrinsic mechanisms (e.g immune checkpoint upregulation) and tumor-extrinsic mechanisms (e.g. recruitment of Tregs and MDSCs
2. The paragraph on PD-1 inhibitors implies that their failure in glioblastoma is due to lower PD-1/PD-L1 expression. Pls add a brief mention of glioblastoma’s highly immunosuppressive microenvironment or poor T-cell infiltration for a more complete explanation.
3. Support the statement that TIM-3 "may offer greater clinical success" with ongoing research or clinical trials that justify its therapeutic potential in glioblastoma.
Tim-3 Biology
1. Add a brief discussion on transcriptional and post-transcriptional regulators of TIM-3 expression, particularly in the glioblastoma microenvironment.
2. This section primarily focuses on TIM-3’s role in T-cell exhaustion, but its functions in NK cells, macrophages, and dendritic cells are not well explained. Add a brief comparison of TIM-3’s function across different immune cell types.
Correlating TIM-3 Expression and Glioblastoma Prognosis
1. Authors have discussed TIM-3 in relation to PD-1, CTLA-4 and CD96. Does TIM-3 co-operate with these checkpoints in glioblastoma progression?
2. Include a brief discussion on whether TIM-3 inhibition alone is sufficient or whether combination therapies with PD-1 inhibitors show better outcomes.
3. Several studies are cited that show TIM-3 expression in relation to macrophages, CD68+ cells and tumor infiltrating lymphocytes. Discuss whether TIM-3+ immune cells contribute to an immunosuppressive microenvironment or are merely a byproduct of tumor progression.
TIM-3 Expression on Immune Cells from Glioblastoma Patients
1. There are contradictory results regarding TIM-3 expression in peripheral blood T cells from glioblastoma patients. Include a brief discussion explaining why different studies might report conflicting results (like differences in patient cohorts, experimental techniques or tumor microenvironment variations).
2. Ensure consistency in terminologies such as glioblastoma vs. glioma, TIM-3+ vs. TIM-3 positive.
TIM-3 Expression on Glioblastoma Cells
1. The discussion of Kim et al.'s findings (TIM-3 promoting IFNγ expression in glial cells) contradicts the generally accepted view of TIM-3 as an immunosuppressive checkpoint. Include a paragraph exploring possible explanations for these contradictions, such as TIM-3 may have context-dependent functions, being pro-tumorigenic in glioblastoma cells but immunoactivating in glial cells. Different experimental conditions (e.g in vivo vs. in vitro models) may account for the observed discrepancies.
Targeting TIM-3 in Glioblastoma
The Kim et al. (2017) study reported that anti-TIM-3 monotherapy had no significant survival benefit whereas other studies showed modest but significant survival improvement. Provide a discussion on why different studies may yield inconsistent results.
Conclusion
The conclusion effectively summarizes the key findings of the review, emphasizing the potential of TIM-3 as a therapeutic target in glioblastoma.
Include three to four key takeaways from the review such as:
-TIM-3 is highly expressed on tumor-infiltrating lymphocytes and glioblastoma cells correlating with worse prognosis.
-TIM-3 blockade enhances T-cell function and reduces glioblastoma progression in preclinical models.
-Combination therapies targeting TIM-3 + PD-1 or TIM-3 + CTLA-4 show more promising results than monotherapies.
-TIM-3 expression can be modulated by chemotherapy (e.g., temozolomide) which may have implications for clinical strategies.
Author Response
Cells Response to Reviewer comments.
Manuscript ID: cells-3476035
Title: The Role of TIM-3 in Glioblastoma Progression.
Authors: Farah Ahmady, Amit Sharma, Adrian Achuthan, George Kannourakis, Rodney B. Luwor*
Reviewer 1:
Overview comment: This review provides an in-depth discussion of the role of TIM-3 in glioblastoma progression, focusing on its function in immunosuppression and potential therapeutic interventions. The manuscript is well structured and comprehensive. The authors have effectively compiled and critically analyzed existing literature, highlighting gaps in knowledge and future research directions. However, certain aspects require clarification, additional discussion and refinement to enhance the manuscript’s clarity and impact. Below are specific comments and suggestions for improvement.
We thank reviewer 1 for their thorough assessment of our manuscript and we welcome their suggested improvements. We have modified our manuscript in line with their comments as outlined below:
Reviewer 1, Comment 1: Authors have mentioned that Stupp protocol has increased survival by 2.5 months. Briefly explain the potential reason (such as glioblastoma’s resistance mechanisms, invasive nature, or limited drug penetration across the blood-brain barrier).
Our Comment 1.1: We thank the reviewer for our thoughtful suggestion for this addition. We have added the sentence: “This minor improvement may be due to glioblastoma’s highly invasive nature, widespread intra and intertumoral heterogeneity and resistance mechanisms and differential drug penetration across the blood-brain-barrier” to the manuscript for further clarity.
Reviewer 1, Comment 2: Previous brain irradiation is mentioned as a risk factor. A short clarification on whether this refers to prior therapeutic radiation (such as for other tumors) or environmental exposure should be given.
Our Comment 1.2: We have modified this phrase to include “previous irradiation therapy (particularly to the neck and head area),” for further clarification.
Reviewer 1, Comment 3: Authors have discussed glioblastoma’s immunosuppressive properties. Clearly differentiate between tumor-intrinsic mechanisms (e.g immune checkpoint upregulation) and tumor-extrinsic mechanisms (e.g. recruitment of Tregs and MDSCs
Our Comment 1.3: We have included the word “tumor extrinsic” and “intrinsically” into this section to help differentiate the glioblastoma immunosuppressive features.
Reviewer 1, Comment 4: The paragraph on PD-1 inhibitors implies that their failure in glioblastoma is due to lower PD-1/PD-L1 expression. Pls add a brief mention of glioblastoma’s highly immunosuppressive microenvironment or poor T-cell infiltration for a more complete explanation.
Our Comment 1.4: We thank the reviewer for this suggestion. We have added “ along with other factors such as glioblastoma’s highly immunosuppressive microenvironment and/or poor T cell infiltration” to this section to provide a more complete explanation.
Reviewer 1, Comment 5: Support the statement that TIM-3 "may offer greater clinical success" with ongoing research or clinical trials that justify its therapeutic potential in glioblastoma.
Our Comment 1.5: We thank the reviewer for their comment 5. We have stated that TIM-3 may offer greater clinical success as a statement as part of this section of the review introducing the key themes we will discuss in this manuscript. The support for this statement comes with the remainer of the review discussion and thus we feel that adding justification for this statement here will lead to repetition and reduced cohesion of our carefully crafted manuscript. As such we have not changed the revised manuscript.
Reviewer 1, Comment 6: Add a brief discussion on transcriptional and post-transcriptional regulators of TIM-3 expression, particularly in the glioblastoma microenvironment.
Our Comment 1.6: There is no literature on the transcriptional and post-transcriptional regulation of TIM-3 in the glioblastoma setting (besides our most recent paper). As such we have include “Very little is known regarding the transcriptional and epigenetic regulation of TIM-3 in the glioblastoma setting although we have shown that T and NK cells from glioblastoma patients display reduced BAT3 expression potentially allowing for enhanced TIM-3 expression and function [35].” To this section.
Reviewer 1, Comment 7: This section primarily focuses on TIM-3’s role in T-cell exhaustion, but its functions in NK cells, macrophages, and dendritic cells are not well explained. Add a brief comparison of TIM-3’s function across different immune cell types.
Our Comment 1.7: Most of the research into the function and structure of TIM-3 has been performed on T cell subsets. We have therefore highlighted pivotal publications on TIM-3 that mainly focus on T cells. We do however discuss several other immune subtypes in subsequent sections in the context of glioblastoma – the major aim of this review. As such we have not revised our original manuscript based on this comment.
Reviewer 1, Comment 8: Authors have discussed TIM-3 in relation to PD-1, CTLA-4 and CD96. Does TIM-3 co-operate with these checkpoints in glioblastoma progression?
Our Comment 1.8: We have searched through the literature and have not found any evidence of whether TIM-3 co-operates with these other molecules in glioblastoma progeression (although blockade of TIM-3 and PD-1 in combination may be an improved strategy for glioblastoma inhibition – see next comment). As such we have not revised our original manuscript based on this comment.
Reviewer 1, Comment 9: Include a brief discussion on whether TIM-3 inhibition alone is sufficient or whether combination therapies with PD-1 inhibitors show better outcomes.
Our Comment 1.9: We discuss TIM-3 inhibition in the “Targeting TIM-3 in Glioblastoma” section of this review where we discuss the possibility/appropriateness of using TIM-3 inhibitors alone or in combination with other agents. As such we will not modify this section based on the reviewers comment.
Reviewer 1, Comment 10: Several studies are cited that show TIM-3 expression in relation to macrophages, CD68+ cells and tumor infiltrating lymphocytes. Discuss whether TIM-3+ immune cells contribute to an immunosuppressive microenvironment or are merely a byproduct of tumor progression.
Our Comment 1.10: We thank the reviewer for this important comment. In this section we have already noted that “ However, despite obtaining information regarding which proteins are up-regulated in correlation with up-regulation of TIM-3, these studies did not explore whether these markers regulate each other’s expression and the potential pro-tumorigenicity-relevant to this correlative expression.” In the original submission. Hoerver, to improve our review and provide more clarity as outlined by the reviewer, we have also added: “In addition, in the correlative studies described above, it is unclear whether these TIM-3 expressing cells significantly contribute to the immunosuppressive phenotype of glioma/glioblastoma patients or are a byproduct of the tumor progression.” To the revised manuscript.
Reviewer 1, Comment 11: There are contradictory results regarding TIM-3 expression in peripheral blood T cells from glioblastoma patients. Include a brief discussion explaining why different studies might report conflicting results (like differences in patient cohorts, experimental techniques or tumor microenvironment variations).
Our Comment 1.11: We thank the reviewer for pointing out this key comment. We have added “It is difficult to determine the exact reasons for these large discrepancies in findings across these studies. However, potential explanations may include differences in patient cohorts, sample size, the storage conditions and length of samples, experimental technique differences and variable tumor microenvironmental factors.” To our revised manuscript to improve this part of the review.
Reviewer 1, Comment 12: Ensure consistency in terminologies such as glioblastoma vs. glioma, TIM-3+ vs. TIM-3 positive.
Our Comment 1.12: We thank the reviewer for this comment which will ensure consistency across our manuscript and therefore enable our manuscript to be easily read and understood. We have carefully reviewed our manuscript based on the above comment and we are confident that we have used the term glioma vs glioblastoma appropriated based on the articles we have included in this manuscript. Briefly, we have used the term glioma to describe results that have been generated for all grades of glioma whereas we have specifically used the term glioblastoma for reports that have used Grade IV glioma specifically or have been able to delineate the grade IV gliomas from lower grades in these studies (i.e: Grade IV glioma = glioblastoma). We have also made sure that the term TIM-3 positive is not used in our review (except in the references) to reduce any confusion our text may have. The term TIM-3 positivity was maintained in our manuscript as this is the correct term to use when describing TIM-3 staining in flow experiments.
Reviewer 1, Comment 13: The discussion of Kim et al.'s findings (TIM-3 promoting IFNγ expression in glial cells) contradicts the generally accepted view of TIM-3 as an immunosuppressive checkpoint. Include a paragraph exploring possible explanations for these contradictions, such as TIM-3 may have context-dependent functions, being pro-tumorigenic in glioblastoma cells but immunoactivating in glial cells. Different experimental conditions (e.g in vivo vs. in vitro models) may account for the observed discrepancies.
Our Comment 1.13: We thank the reviewer for this comment and agree that this report presents data that is contradictory to other reports discussed in this review. We have added these additional sentences to provide some potential explanation as to why there is this discrepancy in this report as outlined: “We can only speculate that these disparity with the bulk of the current literature may be due to differences in mouse versus human glial behaviour, the time points chosen where initial contact with tumor cells may evoke an immune response but later produce chronic inactivity or a context or cell type specific differential function that is currently not well understood.”
Reviewer 1, Comment 14: The Kim et al. (2017) study reported that anti-TIM-3 monotherapy had no significant survival benefit whereas other studies showed modest but significant survival improvement. Provide a discussion on why different studies may yield inconsistent results.
Our Comment 1.14: Once again, we acknowledge that the reports we review here in this current manuscript contains some inconsistencies. We have provided additional discussion as to potentially why these discrepancies may exist as outlined: “The disparity in efficacy of this anti-TIM-3 antibody across these two studies could be due to differences in the inoculation method, inoculation position, inoculation cell number and scheduling of the anti-TIM-3 antibody administration.”.
Reviewer 1, Comment 15: The conclusion effectively summarizes the key findings of the review, emphasizing the potential of TIM-3 as a therapeutic target in glioblastoma.
Include three to four key takeaways from the review such as:
-TIM-3 is highly expressed on tumor-infiltrating lymphocytes and glioblastoma cells correlating with worse prognosis.
-TIM-3 blockade enhances T-cell function and reduces glioblastoma progression in preclinical models.
-Combination therapies targeting TIM-3 + PD-1 or TIM-3 + CTLA-4 show more promising results than monotherapies.
-TIM-3 expression can be modulated by chemotherapy (e.g., temozolomide) which may have implications for clinical strategies.
Our Comment 1.15: We thank the reviewer for outlining key concluding remarks that we may have overlooked. To improve our conclusion and overall manuscript we have re-written our conclusion to incorporate the above suggestions from the reviewer. These changes can be seen through the track changes document attached accompanying this “addressing the reviewer comments” document.
Reviewer 2 Report
Comments and Suggestions for Authors
The review article by Dr. Ahmady et al., entitled “The Role of TIM-3 in Glioblastoma Progression”, explores the function of the immunoregulatory receptor TIM-3 in glioblastoma progression and prognosis, as well as its expression across various immune cell subtypes. Furthermore, the authors discuss the efficacy of anti-TIM-3 therapeutic strategies in preclinical models, considering their potential clinical applications.
The manuscript is well-written and well-organized, serving as a valuable reference for understanding TIM-3's role in tumor progression. The authors provide an analytical discussion of key aspects relevant to the severity of glioblastoma and current immune cell-based therapeutic strategies. TIM-3’s function is described both under physiological conditions and in the context of glioblastoma. The prognostic significance of TIM-3 expression, either alone or in combination with other markers, is thoroughly examined through a review of the literature, alongside evidence of its ectopic expression in glioblastoma cells. Additionally, preclinical studies evaluating the effects of TIM-3-targeting treatments are comprehensively discussed.
Only minor revisions are necessary:
Line 73: Spell out "NSCLC" for clarity.
Line 95: The semicolon after "TIM-3" is inappropriate; replace it with a comma.
Line 241: The phrase “end of the document for further details on references” is unnecessary and should be removed.
References section: Ensure that the reference list adheres to the journal’s author guidelines.
This review provides a well-structured and insightful analysis of TIM-3 in glioblastoma, making a valuable contribution to the field.
Author Response
Reviewer 2:
Overview comments: The review article by Dr. Ahmady et al., entitled “The Role of TIM-3 in Glioblastoma Progression”, explores the function of the immunoregulatory receptor TIM-3 in glioblastoma progression and prognosis, as well as its expression across various immune cell subtypes. Furthermore, the authors discuss the efficacy of anti-TIM-3 therapeutic strategies in preclinical models, considering their potential clinical applications. The manuscript is well-written and well-organized, serving as a valuable reference for understanding TIM-3's role in tumor progression. The authors provide an analytical discussion of key aspects relevant to the severity of glioblastoma and current immune cell-based therapeutic strategies. TIM-3’s function is described both under physiological conditions and in the context of glioblastoma. The prognostic significance of TIM-3 expression, either alone or in combination with other markers, is thoroughly examined through a review of the literature, alongside evidence of its ectopic expression in glioblastoma cells. Additionally, preclinical studies evaluating the effects of TIM-3-targeting treatments are comprehensively discussed.
We thank reviewer 2 for their thorough assessment of our manuscript and we welcome their suggested improvements. We have modified our manuscript in line with their comments as outlined below:
Reviewer 2, Comment 1: Only minor revisions are necessary: Line 73: Spell out "NSCLC" for clarity.
Our Comment 2.1: We thank the reviewer for this comment. We have adjusted the manuscript as guided by the reviewer.
Reviewer 2, Comment 2: Line 95: The semicolon after "TIM-3" is inappropriate; replace it with a comma.
Our Comment 2.2: We thank the reviewer for this comment. We have adjusted the manuscript as guided by the reviewer.
Reviewer 2, Comment 3: Line 241: The phrase “end of the document for further details on references” is unnecessary and should be removed.
Our Comment 2.3: We thank the reviewer for this comment. This was an editing mistake on our account where we did not completely delete the automated formatting from the journal. We have adjusted the manuscript to remove this phrase.
Reviewer 2, Comment 4: References section: Ensure that the reference list adheres to the journal’s author guidelines.
Our Comment 2.4: We thank the reviewer for their reminder to adhere to the journal’s correct referencing format. We have double checked our manuscript and the referencing is formatted correctly in accordance with the journal’s guidelines.
Reviewer 2, Comment 5: This review provides a well-structured and insightful analysis of TIM-3 in glioblastoma, making a valuable contribution to the field.
Our Comment 2.5: We thank the reviewer for their thorough review and their very favourable endorsement of our manuscript.
Reviewer 3 Report
Comments and Suggestions for Authors
In their comprehensive review, Ahmady and colleagues summarize the potential role of the immune modulator TIM-3 in glioblastoma progression. This includes expression of TIM-3 in various immune cells in glioblastoma patients, as well as anti-TIM-3 treatment studies in preclinical rodent models, e.g. a triple therapy approach with anti-TIM-3, anti-PD-1, and stereotactic radiosurgery could improve survival, dependent on functional T cells. This review is well written and highly informative.To avoid confusion or potential misleading of the readers, one may consider replacing "astrocytoma" in line 33 with "astrocytic tumor", "glioma", or similar - just to keep astrocytoma and glioblastoma separated.
Author Response
Reviewer 3:
Overview comments: In their comprehensive review, Ahmady and colleagues summarize the potential role of the immune modulator TIM-3 in glioblastoma progression. This includes expression of TIM-3 in various immune cells in glioblastoma patients, as well as anti-TIM-3 treatment studies in preclinical rodent models, e.g. a triple therapy approach with anti-TIM-3, anti-PD-1, and stereotactic radiosurgery could improve survival, dependent on functional T cells. This review is well written and highly informative.
We thank reviewer 3 for their thorough assessment of our manuscript and we welcome their suggested improvements. We have modified our manuscript in line with their comments as outlined below:
Reviewer 3, comment 1: To avoid confusion or potential misleading of the readers, one may consider replacing "astrocytoma" in line 33 with "astrocytic tumor", "glioma", or similar - just to keep astrocytoma and glioblastoma separated.
Our Comment 3.1: We thank the reviewer for this comment. Our sentence “Glioblastoma, classified as a grade IV, isocitrate dehydrogenase (IDH)-wildtype astrocytoma” uses the term astrocytoma as this is how the most current WHO classification of glioblastoma is described with references 1 and 2 (and others) supporting the use of this term. We have not seen glioblastomas described as astrocytic tumors and hence we would like to keep the astrocytoma term in our revised manuscript.